# Achieving clinically optimal balance between accuracy and simplicity of a formula for manual use: Development of a simple formula for estimating liver graft weight with donor anthropometrics

**Nao Ichihara**[1], **Naoya Sato**[2], **Shigeru Marubashi**[2], **Hiroaki Miyata**[1], **Susumu Eguchi**[3,4], **Hideki Ohdan**[4,5], **Koji Umeshita**[4,6], **Mitsukazu Gotoh**[7]*

1 Department of Healthcare Quality Assessment, University of Tokyo, Bunkyo, Tokyo, Japan, 2 Department of Hepato-Biliary-Pancreatic and Transplant Surgery, Fukushima Medical University, Fukushima, Fukushima, Japan, 3 Department of Surgery, Nagasaki University Graduate School of Biomedical Science, Sakamoto, Nagasaki, Japan, 4 Japanese Liver Transplant Society, Suita, Osaka, Japan, 5 Department of Gastroenterological and Transplant Surgery, Applied Life Sciences, Institute of Biomedical and Health Sciences, Hiroshima University, Hiroshima, Hiroshima, Japan, 6 Division of Health Sciences, Osaka University Graduate School of Medicine, Suita, Osaka, Japan, 7 Osaka General Medical Center, Osaka, Osaka, Japan

ʘ These authors contributed equally to this work.
* mgotoh@fmu.ac.jp

## Abstract

In developing a formula for manual use in clinical settings, simplicity is as important as accuracy. Whole-liver (WL) mass is often estimated using demographic and anthropometric information to calculate the standard liver volume or recommended graft volume in liver transplantation. Multiple formulas for estimating WL mass have been reported, including those with multiple independent variables. However, it is unknown whether multivariable models lead to clinically meaningful improvements in accuracy over univariable models. Our goal was to quantitatively define clinically meaningful improvements in accuracy, which justifies an additional independent variable, and to identify an estimation formula for WL graft weight that best balances accuracy and simplicity given the criterion. From the Japanese Liver Transplantation Society registry, which contains data on all liver transplant cases in Japan, 129 WL donor-graft pairs were extracted. Among the candidate models, those with the smallest cross-validation (CV) root-mean-square error (RMSE) were selected, penalizing model complexity by requiring more complex models to yield a $\geq$5% decrease in CV RMSE. The winning model by voting with random subsets was fitted to the entire dataset to obtain the final formula. External validity was assessed using CV. A simple univariable linear regression formula using body weight (BW) was obtained as follows: WL graft weight [g] = 14.8 × BW [kg] + 439.2. The CV RMSE (g) and coefficient of determination ($R^2$) were 195.2 and 0.548, respectively. In summary, in the development of a simple formula for manually estimating WL weight using demographic and anthropometric variables, a clinically acceptable trade-off

**Data Availability Statement:** The minimal data set for conclusion are within Supporting information files.

**Funding:** This work was supported with grants from the Japan Agency for Medical Research and Development and from Japan Society for the Promotion of Science (19K09111); MG received the grant.

**Competing interests:** The authors have declared that no competing interests exist.

between accuracy and simplicity was quantitatively defined, and the best model was selected using this criterion. A univariable linear model using BW achieved a clinically optimal balance between simplicity and accuracy, while one using body surface area performed similarly.

## Introduction

### Importance of simplicity in developing formulas for manual use

Despite recent advances in machine learning, there are still some clinical areas in which complex algorithms fail to provide clinically meaningful gain of accuracy compared to relatively simple formulas because of the limited availability of data and high variability of the subject, leaving traditional simple estimation/prediction formulas yet to be replaced. When developing a formula in such areas, in addition to predictive accuracy, simplicity and ease of use are major concerns because they are often used manually, and the complexity of the model might limit its use or lead to errors.

### Improving and assessing the external validity of estimation models

For an estimation formula to be of practical value, it is important to ensure that such a formula provides reasonably accurate estimations for unseen samples from a defined population. In other words, external validity should be maximized and confirmed.

Overfitting is a term that describes one of the potential issues in selecting a more complex model. It occurs when a more complex model fails to perform better in an unseen sample from a defined population compared to a simpler model, whereas it performs better in a training sample from the same population. There are established techniques for improving external validity and avoiding overfitting, including cross-validation (CV) and bootstrapping, and these are also used to assess the model's external validity.

### Beyond avoiding overfitting: Seeking practically optimal balance between simplicity and accuracy

Even in the absence of overfitting, the higher complexity of a formula may only lead to marginal rather than practically meaningful improvements in accuracy. In this study, this situation is referred to as *superfitting*.

While there are some widely used statistical measures and methods for balancing the complexity and accuracy of estimation models, including adjusted $R^2$, they do not reflect domain knowledge or help avoid *superfitting*. A mathematical criterion for avoiding *superfitting* needs to reflect its context and purpose. In the simplest form, this involves three steps: (1) defining the measure of model complexity, (2) defining the measure of model accuracy, and (3) defining acceptable trade-offs between these measures. To the best of our knowledge, no such mathematical criterion has been reported in clinical research or any other areas.

### Model accuracy measure for avoiding *superfitting*: RMSE

In applied studies including clinical medicine, the coefficient of determination ($R^2$), defined as ratio of the variance explained by the model to the total variance, is often used as the primary measure of accuracy. While $R^2$ enables a comparison of accuracy between models whose dependent variables have different variances and scales (such as grams and milliliters, each representing weight and volume), it does not convey a difference in accuracy on a clinically

meaningful scale. This makes $R^2$ unsuitable when an optimal balance between the accuracy and simplicity of a model is sought.

The root-mean-square error (RMSE) is, as its name describes, a summary measure of residuals, or "error," in the dimension of the measurements, not as a ratio. Because of this, RMSE allows clinicians to judge the pragmatic significance of accuracy differences between models. Using RMSE for assessing model accuracy enables an objective definition of a clinically acceptable trade-off between the simplicity and accuracy of an estimation formula.

## Estimation of WL graft mass using demographic and anthropometric variables

Estimation of WL mass using demographic and anthropometric information is important in liver transplantation, as it is used to calculate the standard liver volume (SLV) and the recommended graft mass for a recipient. Several such estimation formulas have been developed using parameters such as body weight (BW), body surface area (BSA), body height (BH), age, and sex (Table 1) [1–13]. However, it is not established which independent variable or combination of variables serves best for this purpose.

Some of the previously reported estimation formulas for WL graft mass have multiple independent variables, and some have even more complex structures than simple linear formulas (Table 1). However, cross-validation was not employed for model selection in any of the previous reports. Also, most of the previous reports on multivariable and complex-structured estimation formulas did not even compare the performance of their proposed models against simpler alternatives. This suggests that the previously reported formulas with multiple independent variables and those with relatively complex structures might suffer from overfitting. Such overfitting, by definition, might manifest even in samples similar to their training samples, for example, one in the same country with a similar "racial" composition.

Also, the concern for *superfitting* was not addressed in any of the previous reports, all of which used the $R^2$ as the primary measure of accuracy. While some of them claimed superior accuracy of the proposed multivariable and complex formulas over previously reported simpler ones, the nature of $R^2$ indicates that, even in the absence of overfitting, such formulas might not yield clinically meaningful improvements in accuracy over simpler alternatives, that is, they might be *superfitting*.

In this study, we aimed to develop a formula for manually estimating the weight of WL grafts from adult donors using donor anthropometric and demographic variables. To achieve a clinically optimal balance between accuracy and simplicity of the model, that is, to avoid overfitting and *superfitting*, an objectively defined criterion based on RMSE was implemented in the model selection process.

## Materials and methods

### Japanese liver transplantation society database

The study protocol was approved by the project committee of the Japanese Liver Transplantation Society (JLTS) and the Institutional Review Board of Osaka General Medical Center in accordance with the Declaration of Helsinki.

We used the Japanese Liver Transplantation Society (JLTS) database for information on liver transplant donors and grafts. The JLTS database is a registry of all liver transplant cases in Japan, including both living and deceased donors, operated by JLTS since 2012. The registry includes donor information such as type (living or deceased), age, sex, BW, BH, ABO and Rh blood types, and graft weight. The minimal data set for conclusion are within S1 Data.

**Table 1. Previously reported estimation formulas for whole liver weight or volume and their fit to the current dataset.**

| Authors | Year Country | 1: Formula 2: Number of cases 3: Under 18 years old: | Mean liver mass | Measurement | In-sample $R^2$ | Validation with independent samples | Current cohort | |
|---|---|---|---|---|---|---|---|---|
| | | | | | | | $R^2$ | RMSE |
| Prediction of graft weight (g) | | | | | | | | |
| DeLand and North | 1968 United States | 1: $1020 \times BSAd - 220$ 2: 550 3: not described | 1250g | Autopsy | NA | none | 0.26 | 244.8 |
| Heinemann et al. | 1999 Germany | 1: $1072.8 \times BSAd - 345.7$ 2: 1332 3: not described | Not described | Autopsy | 0.3 | none | 0.36 | 228.2 |
| Yoshizumi et al. | 2003 United States | 1: $772 \times BSAm$ 2: 1413 3: included | 1438.0g (male) 1290.1g (female) | Back table[‡] | 0.73 | none | 0.41 | 217.5 |
| Yu et al. | 2004 Korea | 1: $21.585 \times (BW^{0.7322}) \times (BH^{0.225})$ 2: 652 3: included | 1396.0 ± SD: 377.9 g | Autopsy | 0.59 | none | 0.46 | 208.1 |
| Choukèr et al. | 2004 Germany | 1: $F_1 = 452 + (16.34 \times BW) + (11.85 \times age) - (166 \times sex)$ for (age $>=$ 16 and $<=$ age 50 years) $F_2 = 1390 + (15.94 \times BW) - (12.86 \times age)$ for (age $>=$ 51 and age $<=$ 70 years) 2: 728 3: included | 1664.4 ± SD:478.41 g | Autopsy | 0.38 ($F_1$) 0.35 ($F_2$) | none | -1.76 | 470.4 |
| Current study | 2020 Japan | 1. $14.8 \times BW + 439.2$ 2. 129 3: not included | 1297 (range:1144–1535) g | After perfusion | 0.49 | cross validation | 0.470 | 206.8 |
| Prediction of estimated graft volume by CT volumetry (ml) | | | | | | | | |
| Urata et al. | 1995 Japan | 1: $706.2 \times BSAd + 2.4$ 2: 96 3: included (65 pediatric subjects) | 764 ±SD:380 ml | NA | 0.96 | none | 0.05 | 276.6 |
| Lin et al. | 1998 Taiwan | 1: $13 \times BH + 12 \times BW - 1530$ 2: 33 3: not included | 1291 ± SD:187 ml | NA | 0.83 | none | 0.39 | 221.2 |
| Vauthey et al. | 2002 United states, Switzerland, Belgium | 1: $1267.28 \times BSAm - 794.41$ 2: 292 3: included | 1531(range:649–3558) ml | NA | 0.46 | none | 0.41 | 218.7 |
| Hashimoto et al. | 2006 Japan | 1: $961.3 \times BSAm - 404.8$ 2: 301 3: included | 1196.3 ± SD:221.0 ml | NA | 0.58 | none | 0.19 | 255.1 |
| Chan et al. | 2006 China | 1: $12.3 \times BW + 51 \times sex + 218$ 2: 159 3: not included | 927.54 ± SD:168.78 ml | NA | 0.48 | none | -1.02 | 407.8 |
| Yuan et al. | 2008 China | 1: $949.7 \times BSAd - 48.3 \times age\_factor[†] -247.4$ 2: 112 3: not included | 1220.1 ± SD:216.1 ml | NA | 0.44 | prospectively evaluated using 63 living donors | 0.31 | 236.5 |
| Fu-Gui et al. | 2009 China | 1: $11.508 \times BW + 334.024$ 2: 115 3: not included | 1053.1 ± SD:167.6 ml | NA | 0.36 | none | -0.71 | 372.2 |

(*Continued*)

**Table 1.** (Continued)

| Authors | Year Country | 1: Formula 2: Number of cases 3: Under 18 years old: | Mean liver mass | Measurement | In-sample R² | Validation with independent samples | Current cohort | |
|---------|--------------|-----------------------------------------------------|-----------------|-------------|--------------|-------------------------------------|----------------|---|
| | | | | | | | R² | RMSE |
| Poovathumkadavil et.al | 2010 Saudi Arabia | 1: $12.26 \times \mathbf{BW} + 555.65$ 2: 351 3: included | 1435.0 ± SD:315.4 ml | NA | 0.37 | none | 0.46 | 209.7 |

Abbreviations: R2, coefficient of determination; RMSE, root-mean-square error; BH, body height; BW, body weight; BSAd, body surface area by DuBois formula; BSAm; body surface area by Mosteller formula; CT, computed tomography; and SVL, standard liver volume.

Year, the year of publication; Country, the primary country of data collection; Number of cases, number of cases used for selecting and fitting the model; Measurement, method of measuring liver weight or volume; In-sample R2: reported coefficient of determination (R2) against the samples with which the model was selected and fitted, Validation with independent samples; validation of the model's external validity using dataset independent from the one used for selecting and fitting the model, Current cohort R2 and RMSE; accuracy measures of previously reported prediction models against the current WL dataset and those of the current WL model estimated in the "outer" cross validation for model performance evaluation. Note that accuracy measures of previously reported prediction models against the current WL dataset and these of the current model estimated in the cross validation cannot be directly compared.

‡Graft weight was measured just after the back-table procedure

†Age factor; age was counted as 1 for those <40, 2 if 41–60, and 3 if >60 years old

## Analysis cohorts and variables

The process of creating the analysis cohort is illustrated in Fig 1. We used only data of while liver graft from deceased donor for the current study. Donor features (age, sex, BW, BH, and two estimations of BSA as follows) were examined as candidate independent variables.

BSA was calculated according to the DuBois and DuBois formula: [14] $BSA[m^2] = BW[kg]^{0.425} \times BH[cm]^{0.725} \times 0.007184$ and the Mosteller formula: [15] $BSA[m^2] = \sqrt{\frac{BH[cm] \times BW[kg]}{3600}}$.

## Selecting a model through "inner" CV

The weight of the liver grafts was used as the dependent variable for each linear regression model using any of the above candidate independent variables or their combinations (Fig 2). The root-mean-square error (RMSE) of the candidate models was averaged through "inner" CV (number of folds = 10, number of repetitions = 10), and was used as the criterion for model selection. More complex models (bivariate models) were required to yield a mean CV RMSE smaller than their simpler alternatives (univariable models) by ≥5% ("5% RMSE rule"). A "vote" was given to the selected model.

Total of 100 "votes" were collected, each of which represented results of a cycle of "inner" CV with a different subset. The candidate model with the largest number of "votes" was selected as the final model. The final model was fitted to the entire data to obtain the intercept and coefficient of the formula.

## CV for model evaluation ("outer" CV)

"Outer" CV (number of folds = 10 and number of repetitions = 10) was employed to estimate the external validity of the final model. In addition to RMSE and $R^2$, the frequency of each model selected through this "outer" CV was presented as a description of sampling-related variance of model selection. The combination of "inner" and "outer" CV (nested CV) is schematically described in Fig 3.

The "outer" cross-validation: The "outer" cross-validation is for estimating accuracy of the final model on unseen sample from the same population. As illustrated above, the entire model selection and fitting process, including the "inner" cross-validation, was repeated with

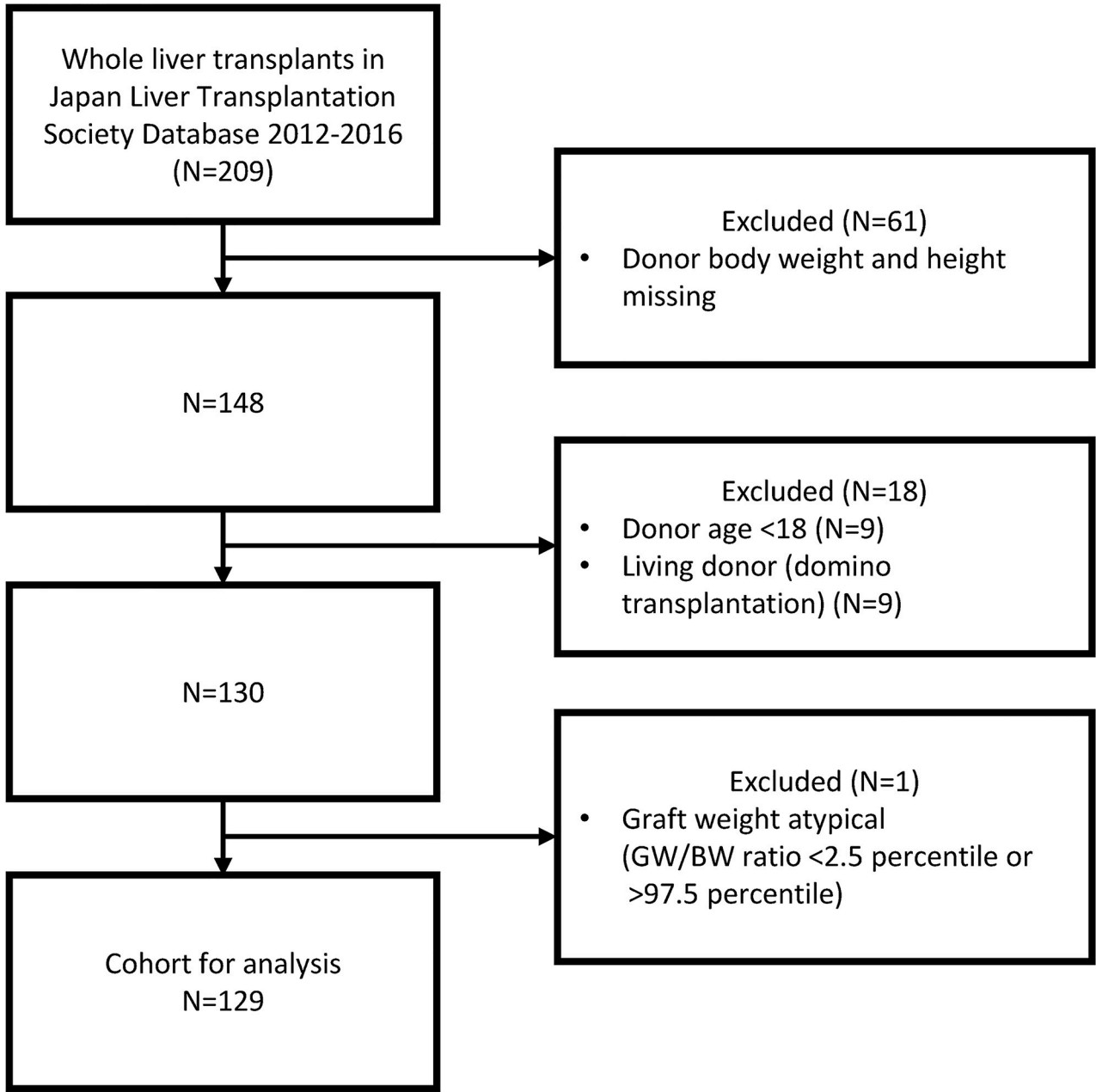

**Fig 1. Selection of donor/graft pairs.** There were no missing values in age, sex, or graft weight.

different subsets of the data (training data), and their accuracy was evaluated using the spare data (test data). The model's accuracy measures are not biased when the total sample size is large enough, and the difference in sample size between the entire dataset and training dataset in the "outer" cross-validation is not critical.

Distribution of continuous variables was reported as medians with interquartile ranges, and that of categorical variables was expressed as numbers and prevalence rates. R software version 3.6.2 (R Foundation for Statistical Computing, Vienna, Austria, 2019) was used for statistical analysis.

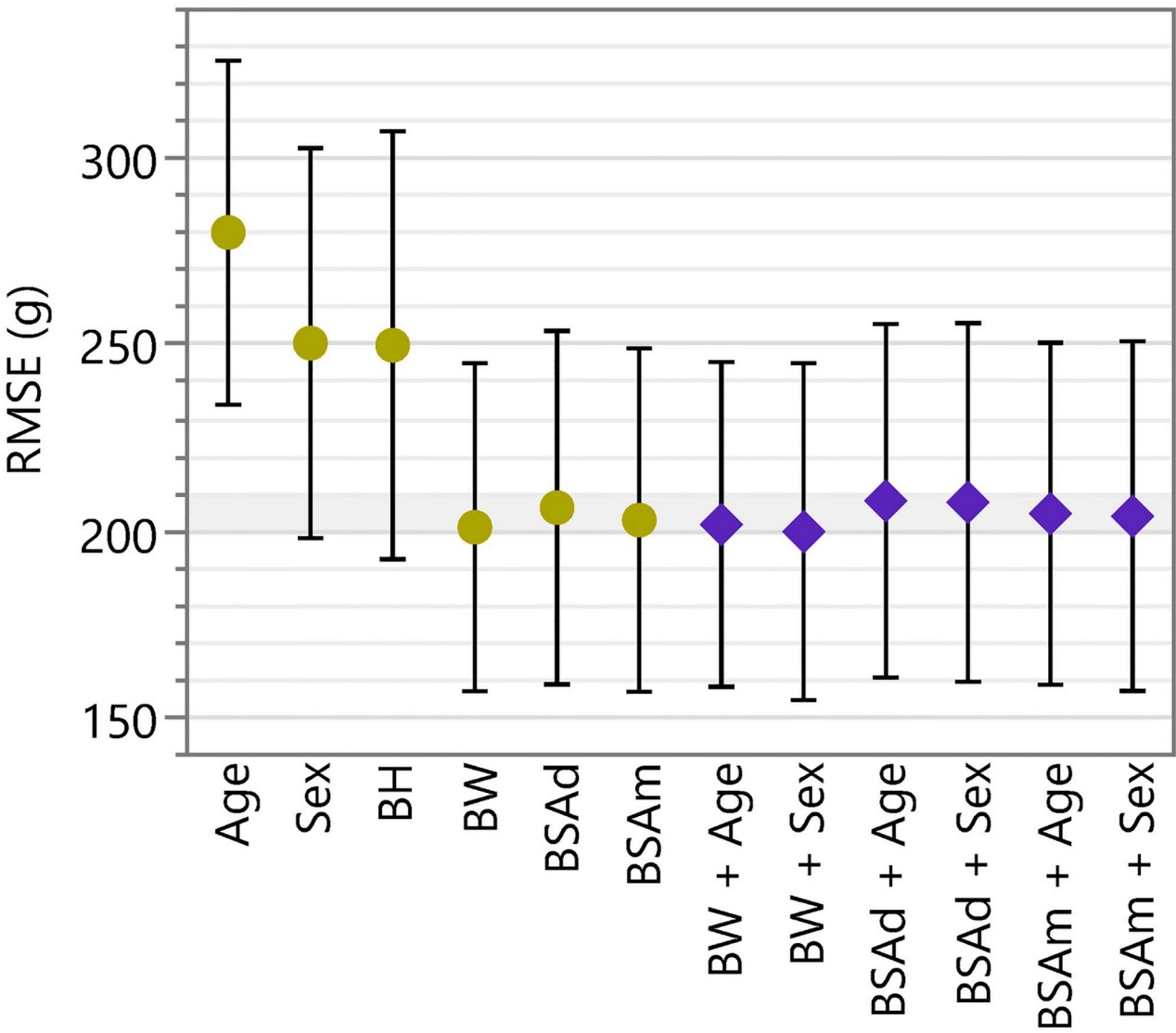

**Fig 2. RMSE of the candidate models.** The y-axis represents the cross-validation RMSE of the candidate models, i.e., (combination of) candidate independent variable(s). The dots represent the mean, and the bars represent the standard deviation, both across iterations in the "inner" CV. The lower border of the gray band represents the smallest mean among all candidate models. The higher border of the band represents this value multiplied by 1.05. The color and shape of the markers represent the complexity, i.e., the number of independent variables, of the models. RMSE, root-mean-square error; indep vars, independent variables; BW, body weight; BH, body height; BSAd, body surface area (Du Bois and Du Bois), BSAm: body surface area (Mosteller).

## Results

### Donor and graft characteristics

Among the liver transplant cases in the JLTS database between 2012 and 2016 (n = 2,181), 129 pairs of WL donors and grafts were included in the analysis (Fig 1). The characteristics of the study cohort are summarized in Table 2.

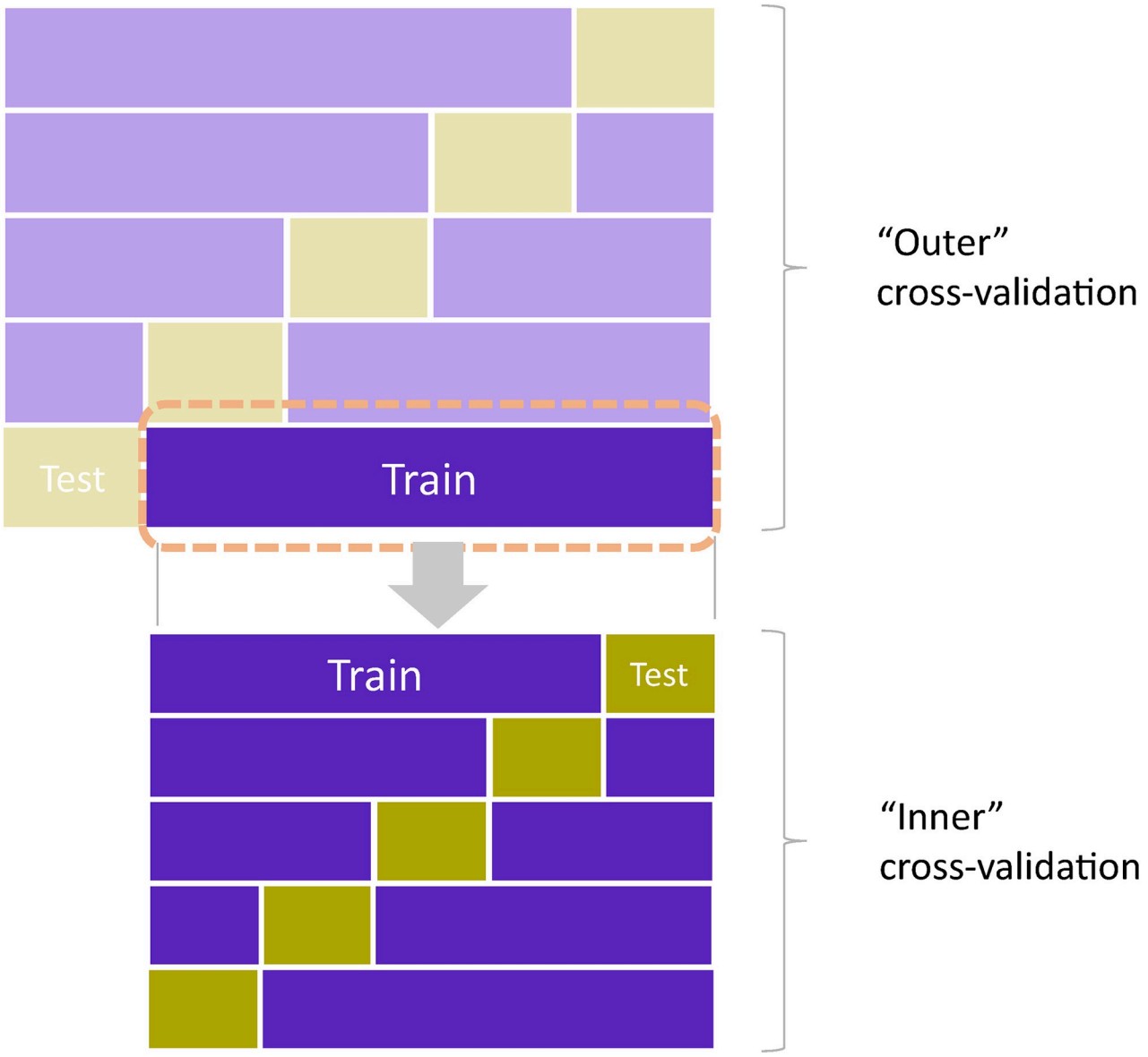

**Fig 3. Schematic description of nested cross-validation.** The "inner" cross-validation: The "inner" cross-validation is for model selection based on their accuracy with unseen data. Here, the models are repeatedly fitted to different random subsets (training data), and their accuracy is evaluated with the data not used for fitting (test data). The model's accuracy with the test data was averaged through iterations and used for model selection.

## Performance of the candidate models

The distribution of the RMSE of the candidate models measured in the "inner" CV is illustrated in Fig 2. Similarly, the distribution of $R^2$ is illustrated in S2 Fig. Models with multiple independent variables received no "votes" based on the "5% RMSE rule," and models using BW, Du Bois and Du Bois' BSA, and Mosteller's BSA received "votes." (The distribution of "votes" is not shown).

**Table 2. Characteristics of donors and liver grafts.**

| Variable | |
|---|---|
| N | 129 |
| Donor features | |
| Age (years) | 46 (37–56) |
| Sex (female) | 56 (43.4%) |
| Height (cm) | 165 (158–170) |
| Weight (kg) | 60 (51–69) |
| Body surface area | |
| Du Bois | 1.67 (1.52–1.79) |
| Mosteller | 1.67 (1.52–1.80) |
| Graft weight (g) | 1297 (1144–1535) |

The continuous variables were reported as medians and interquartile ranges, and the categorical variable (sex) was reported as prevalence rate. Body surface area was calculated according to the formulas reported by Du Bois, or Mosteller formula, as follows. Du Bois formula = BW (kg) $^{0.425}$ × BH (cm) $^{0.725}$ × 0.007184. Mosteller formula = $\sqrt{\frac{BH\ (cm)\ \times\ BW\ (kg)}{3600}}$.

## Final formula and measures of its estimated external validity

Table 3 summarizes the final fitted formulas, its in-sample fit measures, and the results of CV for model performance evaluation ("outer" CV). In-sample fit measures represent the apparent accuracy of the model on the data used to select and fit them. These do not represent the accuracy of the model on unseen data and are presented primarily for comparison with previous studies. The univariable model using BW was finally selected, whereas the univariable model using Mosteller's BSA was selected in 5% of iterations in the "outer" CV.

The final fitted formula, with its CV RMSE and $R^2$, is as follows:

$$\text{WL graft weight [g]} = 14.8 \times \text{BW [kg]} + 439.2,$$

$$\text{RMSE; } 195.2 \text{ [g], } R^2; \ 0.548$$

**Table 3. Final fitted model with results of cross validation.**

| Final fitted model | | | |
|---|---|---|---|
| Graft weight [g] = 14.8 × BW [kg] + 439.2 | | | |
| In-sample fit | RMSE (g) | | 202.9 |
| | $R^2$ | | 0.490 |
| Cross validation | Candidate variable | BW | BSAm |
| | Frequency of being selected (%) | 95% | 5% |
| | Accuracy | RMSE (g) | 195.2 |
| | | $R^2$ | 0.548 |

Abbreviations: BW; body weight, BSAm; body surface area (Mosteller formula), RMSE; root-mean-square error, $R^2$: coefficient of determination.

In-sample fit measures represent degree of apparent accuracy of the models on the data used for fitting them. These were presented primarily for comparison with previous studies. Cross validation results are each candidate model's frequency of being selected as the best model and its accuracy measures against independent samples in the "outer" cross validation for model evaluation. Linear regression formulae took the following form: (graft weight) = $a + b_1 x_1 + b_2 x_2$ ($x_1$, $x_2$: independent variables listed in the table. $a$: intercept. $b_1$: $b_2$ coefficients.).

The predictive performance of the formula was also confirmed using an actual *vs.* estimated plot (Fig 4).

Given these results, we concluded that a simple univariable linear formula using donor BW achieves a clinically optimal balance between accuracy and simplicity in estimating WL graft weight among linear formulas using donor demographic and anthropometric variables. The use of two or more variables did not allow a clinically meaningful gain of accuracy, defined as an approximately 5% reduction in RMSE (the "5% RMSE rule").

**Previously reported estimation formulas and their fit to the current dataset**

Table 1 summarizes previously reported estimation formulas for WL graft weight or volume.

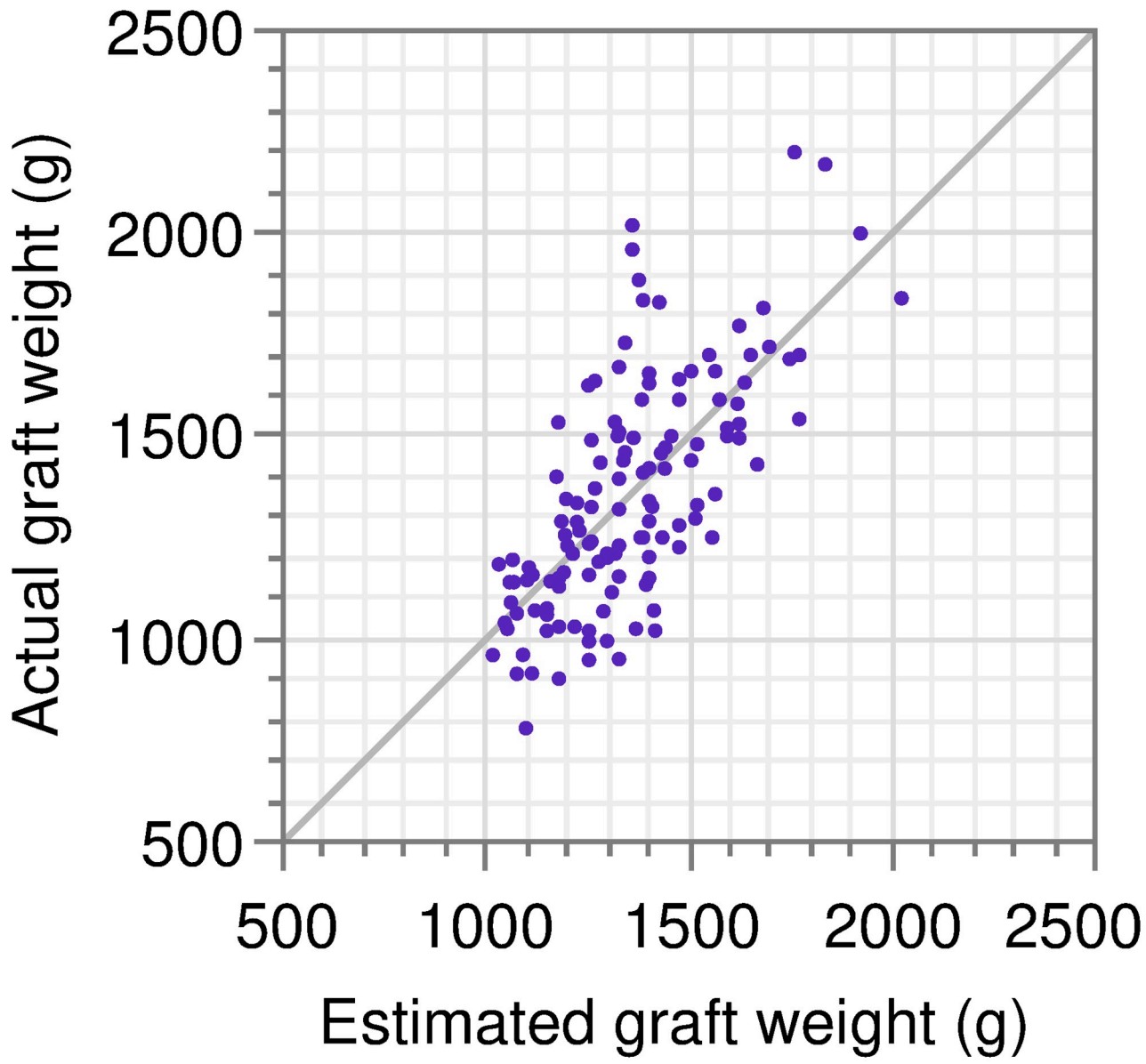

**Fig 4. Actual *vs.* predicted plot of the final fitted model.** Scatter plot between actual graft weight in the current cohort and estimated graft weight with the final fitted models using BW. The 45° line through the origin represents perfect precision.

The fit of these estimation formulas to the current population was examined, while disregarding differences between populations in previous studies (Western *vs.* Asian, and individuals with no liver diseases or liver donor candidates *vs.* actual liver donors), scale (weight *vs.* volume), as well as measurement methods (back table *vs.* autopsy *vs.* CT). It was assumed that the measured volume (mL) was equal to the weight (g).

The fit of these formulas to the current population is summarized with $R^2$ (Table 1 and Fig 5), RMSE (Table 1 and S1 Fig), and actual *vs.* estimated plots (Fig 6).

The previously reported formulas for estimating liver mass showed variable accuracy, as represented by $R^2$ and RMSE, in the current cohort.

## Discussion

### Summary of findings

To the best of our knowledge, this is the first study to quantitatively define the clinically optimal balance between accuracy and simplicity of an estimation formula for manual use, and to empirically select a formula that best meets such a criterion. The context was the estimation of WL graft weight from adult donors using demographic and anthropometric variables. To balance accuracy and simplicity, it was assumed that incorporating two or more variables should be justified with an approximately ≥5% decrease in RMSE (the "5% RMSE rule"). The conclusion was the selection of a univariable linear formula using BW.

### Incorporating clinical knowledge in model development

This is the first study in which an estimation formula for WL graft mass was developed using RMSE, a measure of model accuracy in a clinically meaningful unit (gram), combined with a criterion that reflects clinical knowledge on the optimal balance between the accuracy and complexity of estimation formulas. It was shown that balancing model complexity and accuracy based on a clinically defined criterion (the "5% RMSE rule") selects a simple univariable formula. This is aligned with the apparent lack of correlation between the number of independent variables of previously reported formulas and their RMSE or $R^2$ in the current population (Fig 5 and S1 Fig).

In this study, the number of independent variables was used as a measure of model complexity, RMSE was used as a measure of model performance, and an acceptable trade-off was defined as a 5% reduction in RMSE for one additional independent variable. In the language of machine learning, the cost function to be minimized was defined as $C = R\,(1 + 0.05(N\text{-}1))$, where $C$ represents the cost function, $R$ represents RMSE(g), and $N$ represents the number of independent variables. Despite potential disagreement on the clinical criterion that warrants exploration of other clinically viable criteria, incorporating clinical knowledge in the model development process represents an important principle that could improve the pragmatic implications of a broad range of research involving the development of prediction/estimation formulas for manual use.

### Review of previously reported formulas for estimating liver mass

See S1 Text. Review of previously reported formulas for estimating liver mass.

### Variable selection in liver mass estimation

See S2 Text. Variable selection in liver mass estimation.

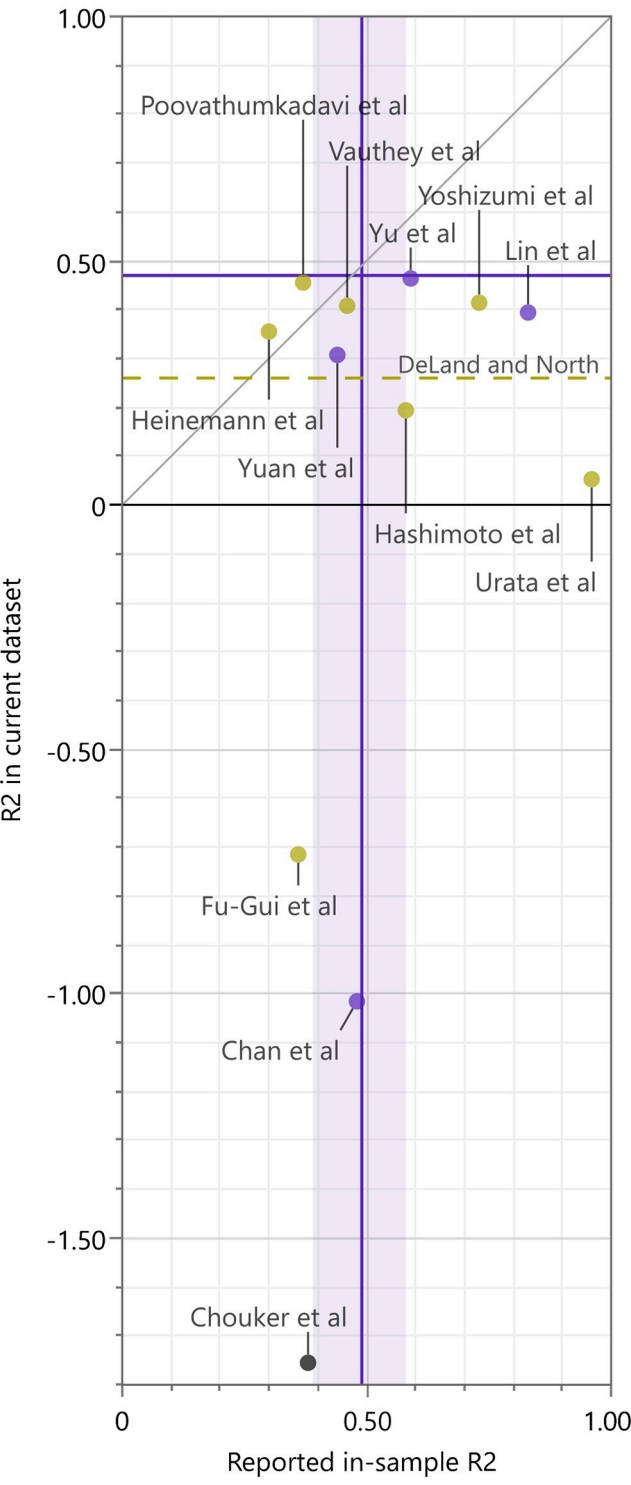

**Fig 5. $R^2$ of the previously reported models.** The y-axis represents the coefficient of determination ($R^2$) of each previously reported prediction model measured in the current whole-liver (WL) cohort. The x-axis represents the reported in-sample $R^2$ of previously reported prediction models, i.e., $R^2$ measured against the dataset with which the model was selected and fitted. The blue purple horizontal line represents cross-validation $R^2$ of the current model. The blue purple vertical line, with a band, represents the in-sample $R^2$ of the current model, i.e., $R^2$ of the current model measured against the current dataset, with its bootstrap 95% range. The dark yellow dashed horizontal line represents the $R^2$ from the prediction model of DeLand and North in the current dataset (these authors did not report in-sample $R^2$). The color of the dots represents the number of independent variables used in each prediction model. Dark yellow: 1; blue purple: 2; black: 3 or more. $R^2$, coefficient of determination.

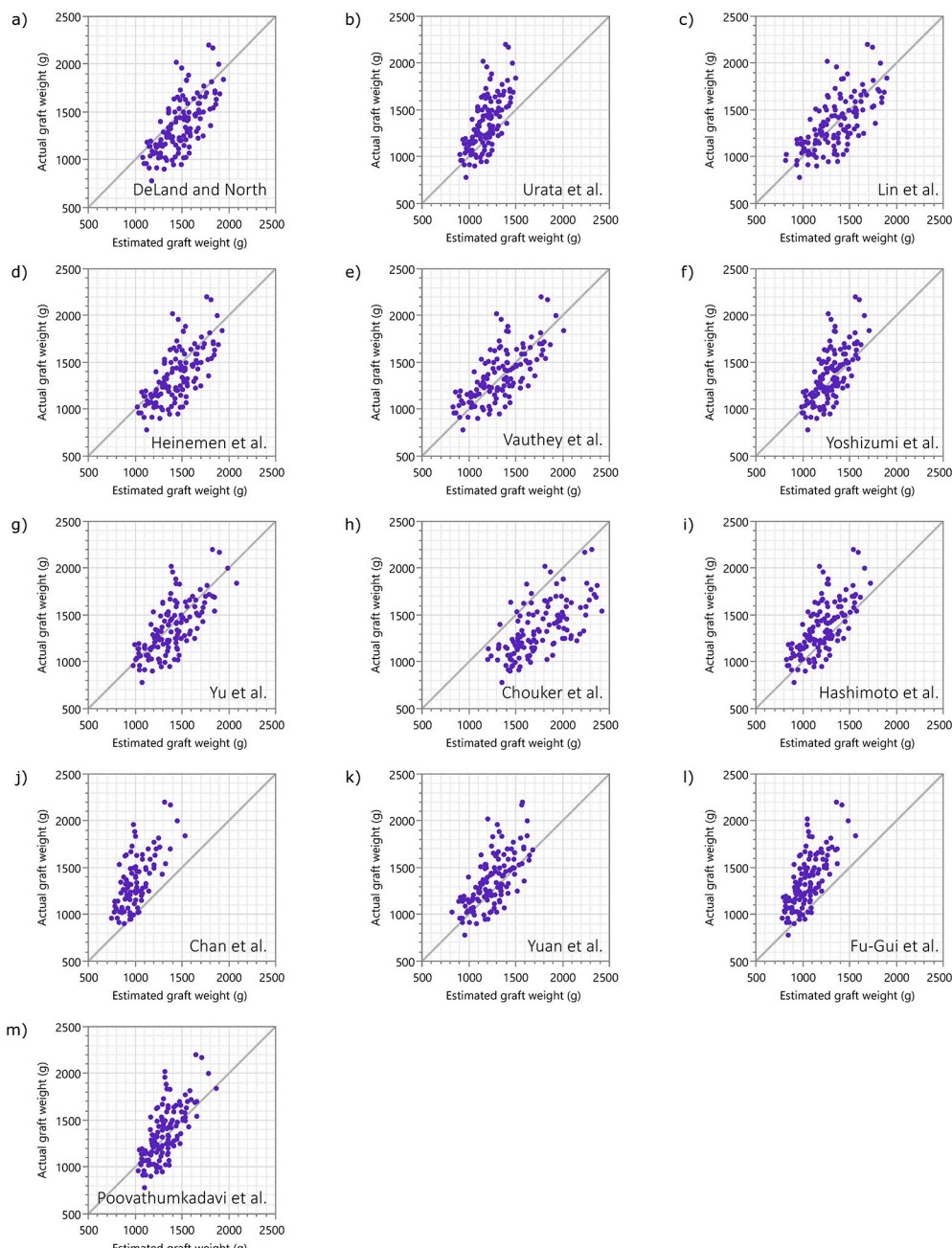

**Fig 6. Actual vs. estimated plots of the previously reported models.** Scatter plots between the actual graft weight in the current cohort and the estimated graft weight using previously reported formulas. The 45˚ line through the origin represents perfect precision.

## Limitations

It should be noted that, even when the goal was to estimate liver graft weight in a similar population as in this study, a more complex model could be selected if a substantially larger sample was used for analysis. It is also worth noting that the definition of the optimal balance between simplicity and accuracy, reflected in the cost function, involves not only clinical but also subjective judgment, and it is worthwhile to explore other potential criteria.

Regarding the applicability of the formula developed here to different populations, similar miscalibration as observed with the previously reported univariable linear formulas in the current dataset, as summarized in by Table 1, Figs 5, 6, and S1 Fig, is anticipated. Thus, developing an estimation formula optimized in a specific population may still be justified in future, unless a "universal" formula is developed and proven, which will require a far more diverse samples. In absence of such a "universal" formula, however, the authors believe the current study will provide guidance on model selection for developing a formula in a specific population using similar volume of data.

## Conclusion

When a formula for manual use in a clinical setting is selected and fitted, a clinically optimal balance between accuracy and simplicity can be achieved using an objective criterion. Using such a criterion, in WL graft weight estimation using demographic and anthropometric variables, a univariable linear formula using BW was found to be optimal, balancing simplicity and accuracy.

## Supporting information

**S1 Fig. Fit of previously reported formulas to current dataset.** RMSE of the previously reported formulas measured in the current dataset. The gray horizontal line represents the cross-validation RMSE of the current formula. The color of the bars represents the number of independent variables used in each prediction model. Dark yellow: 1; blue purple: 2; black: 3 or more. The models are grouped by the method of liver mass measurement employed, which is presented at the bottom. RMSE, root-mean-square error.
(TIF)

**S2 Fig. $R^2$ of candidate models.** The y-axis represents the coefficient of determination ($R^2$) of the candidate models, *i.e.*, the combination of candidate independent variables. The markers represent the mean, and the bars represent the standard deviation. The color and shape of the markers represent the complexity, *i.e.*, the number of independent variables, of the models. Dark yellow: 1; blue purple: 2. $R^2$, coefficient of determination; indep vars, independent variables; BW, body weight; BH, body height; BSAd, body surface area (Du Bois and Du Bois); BSAm, body surface area (Mosteller).
(TIF)

**S1 Text. Review of previously reported formulas for estimating liver mass** [2, 5, 7, 10, 13, 22, 23].
(DOCX)

**S2 Text. Variable selection in liver mass estimation** [7, 5, 9, 11, 16–21].
(DOCX)

**S1 Data. Minimal dataset for reproducing the analysis.**
(XLSX)

## Acknowledgments

The authors thank Dr. Shinji Uemoto (immediate past president, Japanese Liver Transplant Society), Dr. Yukihiro Inomata (Registration Committee, Japanese Liver Transplant Society), and Dr. Hiroyuki Furukawa (Project Committee, Japanese Liver Transplant Society) for their outstanding contributions to the present study.

## Author Contributions

**Conceptualization:** Nao Ichihara.

**Data curation:** Susumu Eguchi, Hideki Ohdan, Koji Umeshita.

**Formal analysis:** Nao Ichihara.

**Funding acquisition:** Mitsukazu Gotoh.

**Project administration:** Mitsukazu Gotoh.

**Supervision:** Shigeru Marubashi, Hiroaki Miyata, Mitsukazu Gotoh.

**Writing – original draft:** Nao Ichihara, Naoya Sato.

**Writing – review & editing:** Nao Ichihara, Naoya Sato, Shigeru Marubashi, Susumu Eguchi, Hideki Ohdan, Koji Umeshita, Mitsukazu Gotoh.

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
