## [Decision Letter · Decision Letter 0]

18 Jul 2022

PONE-D-21-33293Achieving clinically optimal balance between accuracy and simplicity of a formula for manual use: development of a simple formula for estimating liver graft weight with donor anthropometricsPLOS ONE

Dear Dr. Gotoh,

Thank you for submitting your manuscript to PLOS ONE. After careful consideration, we feel that it has merit but does not fully meet PLOS ONE’s publication criteria as it currently stands. Therefore, we invite you to submit a revised version of the manuscript that addresses the points raised during the review process.

We look forward to receiving your revised manuscript.

Kind regards,

Zhen Hua Hu, MD, PhD

Academic Editor

PLOS ONE

Journal Requirements:

2. Please include your tables as part of your main manuscript and remove the individual files. Please note that supplementary tables (should remain/ be uploaded) as separate "supporting information" files.’

3. Please amend your current ethics statement to address the following concerns:

a) Did participants provide their written or verbal informed consent to participate in this study?

5. Thank you for stating the following financial disclosure: "This work was supported with grants from the Japan Agency for Medical Research and Development and from Japan Society for the Promotion of Science (19K09111); MG received the grant."

Please state what role the funders took in the study.  If the funders had no role, please state: "The funders had no role in study design, data collection and analysis, decision to publish, or preparation of the manuscript.

7. Your ethics statement should only appear in the Methods section of your manuscript. If your ethics statement is written in any section besides the Methods, please move it to the Methods section and delete it from any other section. Please ensure that your ethics statement is included in your manuscript, as the ethics statement entered into the online submission form will not be published alongside your manuscript. 

Reviewers' comments:

Reviewer's Responses to Questions

**Comments to the Author**

1. Is the manuscript technically sound, and do the data support the conclusions?

Reviewer #1: Yes

Reviewer #2: Partly

2. Has the statistical analysis been performed appropriately and rigorously? 

Reviewer #1: Yes

Reviewer #2: Yes

3. Have the authors made all data underlying the findings in their manuscript fully available?

Reviewer #1: Yes

Reviewer #2: Yes

4. Is the manuscript presented in an intelligible fashion and written in standard English?

Reviewer #1: Yes

Reviewer #2: Yes

5. Review Comments to the Author

Reviewer #1: Major points:

The authors have generated a formula for liver weight estimates that balanced simplicity with accuracy. More complex formulas (Yu, Chouker, Lin, and Chan) may be affected by overfitting and super fitting. It is critical to justify why a new formula is needed over already existing simple, linear formulas (DeLand and North, Yoshizumi, Urata, Vauthey, and Hashimoto).

The homogeneity of the study population may decrease the generalizability. This also may have contributed to the decrease in “in-sample” vs “current cohort” correlation coefficients for previous studies as seen in Table 1. This should be addressed in the discussion section.

Minor points:

Consider using different shapes rather than or in addition to color to differentiate groups in your figures to aid readability for colorblind readers or when printed

Page 4, last sentence of abstract: you use the terms “simple linear model” and “univariable model.” These are considered interchangeable by most lay literature. If both models are truly simple linear, it would be less confusing to the reader to use consistent terminology. If there are differences in the formulas, then please add the univariable model using BSA to table 3 for comparison.

Table 1: Study by Yu et al, 2004 includes pediatric data

Figure 1: In the text, please note that you are focused on deceased donors so as to should justify rationale for exclusion of domino transplants as this is still a whole liver

Figure 3: Remove discussion of fitting the final model from your legends

Page 13: concept of votes is unclear and confusing. Consider “models with multiple independent variables were excluded if RMSE did not decrease by more than 5% from univariate models,” or please justify why this is not an accurate statement

Page 17-18: section “R2 and RMSE as measure of model fit” is more appropriate for the introduction

It might be valuable to provide some detail on the RMSE compared to R2

Reviewer #2: The manuscript by Ichihara N et al develops a simple formula to calculate the Whole-liver mass. This parameter is important to establish the size of reduced liver grafts, safety in living liver donors and, increasingly important, the limits of extreme liver resections. The authors arrive at a simple formula after an analysis of 129 donor-recipient pairs in a Japanese transplant database. A simple univariate linear regression formula is achieved, based on a linear relationship with body weight (BW). The authors perform a validation after training and a comparison with other metrics that address the same problem. The formula obtained is simple and easy to apply.

The article is interesting and its methodological design is basically correct. However, the manuscript requires various clarifications:

1. It is not true that the Whole-liver mass calculation problem is better done using simple metrics versus Deep-learning tools. These artificial intelligence tools have shown greater superiority in the field of prediction than statistical tools, although they require a greater inclusion of cases for training and subsequent validation. Although some of these classifiers are black-box models (for example, artificial neural networks) in which the predictor variables are not known, there are other classifiers such as Random Forest, which allow the variables and their relative risk to be known, especially in databases. small data sets, like the one used in this article.

2. The sample size necessary for reliability is not specified. JLTS transplants since 2012 are analyzed to define the analysis cohort.

3. The results are only applicable to the database where the formula was created, since they do not include differential factors such as race, which can influence liver size.

6. PLOS authors have the option to publish the peer review history of their article (what does this mean?). If published, this will include your full peer review and any attached files.

Reviewer #1: No

Reviewer #2: No

---

## [Author Response · Author response to Decision Letter 0]

1 Sep 2022

We appreciated the suggestive comments from the two reviewers that allowed us to refine the manuscript. After carefully reading the comments from the associate editor and the reviewers, we have revised the manuscript extensively. Our detailed point-by-point responses to your comments are in the appended letter. The comments from the reviewers are presented, followed by our response.

【 point-by-point response to the comments from reviewers】

【Comment from Reviewer 1】

The authors have generated a formula for liver weight estimates that balanced simplicity with accuracy. More complex formulas (Yu, Chouker, Lin, and Chan) may be affected by overfitting and super fitting. It is critical to justify why a new formula is needed over already existing simple, linear formulas (DeLand and North, Yoshizumi, Urata, Vauthey, and Hashimoto).

(Our response)

We appreciate the reviewer’s point that we first need to clarify why we need to develop a new estimation formula for whole liver weight while there are multiple published formulas for this purpose. 

The critical reason for this was that (1) none of the previously published formulas have been cross-validated either for model selection or validation so their external validity (within the same population) was questionable and unclear, and (2) there was no comprehensive report on practical value of these formulas in populations different from the one with which they were developed. Thus, we (1) developed a formula with internal and external cross validation so its external validity was maximized and evaluated, and (2) evaluated fit of previously published formulas on our dataset to examine how they perform on a different population.

Although we didn’t explicitly mention in the manuscript, advancement of machine learning (ML) and complex statistics (SC) is an important background for this report. We expect that interest will grow in applying ML/CS tools in this and similar clinical scenarios. In light of such anticipated interest in combining “big data” (BD) and ML/CS methods, the authors believe research areas such as this deserves renewed attention with a focus on both their commonality with and also their uniqueness compared to the mainstream BD/ML/CS research. The authors believe different application areas deserves mathematical/computational methods tailored to their unique real world needs, not only borrowing from the largely commercially-led “mainstream” BD research. 

Although existing reports on such estimation formulas provide fair guidance for future work, we believe some methodological updates are required for this field to live up to the promise of modern science:

1. Measures should be taken to avoid overfitting, e.g., (internal) cross-validation for model selection.

2. Envisioned use case of the estimation formula should be clarified, e.g., manual use (or calculation with a simple electronic calculator), calculation with a feature-rich electronic calculator (“scientific calculator”), computerized calculator that instantly responds to a user’s inquiry, or computerized estimation in settings where the computational load is not a concern.

3. (When the purpose of developing the formula is for manual use) measures should be taken to avoid selecting an excessively complex formula just to gain clinically negligible gain of accuracy, i.e., superfitting. Penalty on model complexity should be enforced as part of the criteria for model selection, i.e., the cost function. 

4. Results of (External) cross validation should be done for estimating external validity of the final fitted formula (in the same population) and its results should be presented.

5. Ideally, performance of the formula in multiple completely different population should be assessed and presented. As this is difficult in most cases, we believe it’s a good practice to assess performance of previously reported formulas in the dataset available to the authors and present such results. Although this could make a separate manuscript, we believe this makes sense to present such results in a manuscript where a new formula is developed, because this is usually the best thing we can do to have an idea of how the newly developed formula performs in a completely different population.

6. The process taken to select the final model should be made clear. At least, the list of candidate models and how they compared against the one finally selected should be presented.

The authors believe these principles also apply to many other areas of applied science, including medicine.

2. The homogeneity of the study population may decrease the generalizability. This also may have contributed to the decrease in “in-sample” vs “current cohort” correlation coefficients for previous studies as seen in Table 1. This should be addressed in the discussion section.

(Our response)

We appreciate the reviewer’s suggestion. 

To clarify, before this study, because there was no report on performance of existing estimation formulas on different populations from where they were developed, nothing was directly known about how such estimation formulas perform in different populations. However, the results here, summarized by Table 1, Fig 5, Fig 6, and S1 Fig, suggest what the reviewer states here is the case. We added the following to the Limitation section.

Regarding the applicability of the formula developed here to different populations, similar miscalibration as observed with previously reported univariable linear formulas in the current dataset, as summarized in by Table 1, Fig 5, Fig 6, and S1 Fig, is anticipated. Thus, developing an estimation formula optimized in a specific population may still be justified in future, unless a “universal” formula is developed and proven, which will require a far more diverse samples. In absence of such a “universal” formula, however, the authors believe the current study will provide guidance on model selection for developing a formula in a specific population using similar volume of data.

3. Consider using different shapes rather than or in addition to color to differentiate groups in your figures to aid readability for colorblind readers or when printed 

(Our response)

We appreciate this comment, and modified all the color-containing figures so they can better be comprehended by the red-green color-blind and also the total color-blind. Colors were selected to provide consistent experience across those affected by color blindness and those who are not. (Now no figures contain red or green.) Specifically, the following changes were made:

• Figure 2 and S2: Modified coloring with varied saturation, varied marker shape, and legends containing color samples.

• Figure 5: Modified coloring with varied saturation.

• Figure S1: Modified coloring with varied saturation, and legends containing color samples.

Although varied maker shapes were not used in Figure 5 for some technical reasons, we believe this is now comprehensible even by those with color-blindness.

4. Page 4, last sentence of abstract: you use the terms “simple linear model” and “univariable model.” These are considered interchangeable by most lay literature. If both models are truly simple linear, it would be less confusing to the reader to use consistent terminology. If there are differences in the formulas, then please add the univariable model using BSA to table 3 for comparison. 

(Our response)

We appreciate this comment and agree with the reviewer regarding this. We made the following change to improve clarity and flow. (The underlines are not present in the manuscript.)

Before change: A simple linear model using BW achieves a clinically optimal balance between simplicity and accuracy, while a univariable model using body surface area performed similarly.

After change: A univariable linear model using BW achieved a clinically optimal balance between simplicity and accuracy, while one using body surface area performed similarly.

We hope the reviewer finds this acceptable.

5. Table 1: Study by Yu et al, 2004 includes pediatric data

(our response)；

Thank you for pointing our mistake. We carefully read the article reported by Yu et al. and found that the study included pediatric data. We changed the Table 1 (please see at page 7 in the revised manuscript).

6. Figure 1: In the text, please note that you are focused on deceased donors so as to should justify rationale for exclusion of domino transplants as this is still a whole liver

(our response)

Thank you for the comment. We added an explanation for the analysis cohort in the text to clarify this issue (please see at 11 page).The added sentence was described below.

“We used only data of whole liver grafts from deceased donors for the current study.”

7. Figure 3: Remove discussion of fitting the final model from your legends

(Our response)

We appreciate this suggestion and admit the need to streamline the legend for S2 Fig. We removed the paragraph describing how the final model is fitted.

8．Page 13: concept of votes is unclear and confusing. Consider “models with multiple independent variables were excluded if RMSE did not decrease by more than 5% from univariate models,” or please justify why this is not an accurate statement

(Our response)

We appreciate this comment, which points to insufficient description of “voting” in this manuscript. We realized that the relationship between the “inner” CV, “voting,” and “outer” CV was not adequately described. 

To clarify, like a typical nested CV with two layers of iterations, one of which is nested within the other, the current algorithm involved three layers of iterations. The inner-most iteration was the “inner” CV, the one in the middle was the cycle for “voting,” and the outer-most iteration was the “outer” CV. 

Thus, for selecting a single “vote,” a cycle of “inner” CV, which encompassed 10 x 10 = 100 iterations of fitting a model and measuring its CV RMSE, was conducted. The “vote” was decided by comparing the mean CV RMSE of each candidate model across this cycle. 

In total, 100 of such “votes” were collected, and the candidate model with the largest number of “votes” were selected as the final model.

For assessing the external validity of the final model, this process was repeated 10 x 10 = 100 times through the “outer” CV cycle.

To clearly yet concisely describe this relationship, we added the following in the Selecting a model through “inner” CV section of the Materials and methods section. (The part with an underline, not present in the manuscript itself, represents the addition for clarification.)

Total of 100 “votes” were collected, each of which represened results of “inner” CV with a different subset. The candidate model with the largest number of “votes” was selected as the final model. The final model was fitted to the entire data to obtain the intercept and coefficient of the formula.

9．Page 17-18: section “R2 and RMSE as measure of model fit” is more appropriate for the introduction. It might be valuable to provide some detail on the RMSE compared to R2

(Our response)

We thank the reviewer for these suggestions. In the previous version, employment of RMSE first appeared in the Methods section without any preceding explanation on the nature of RMSE and purpose of its implementation, which were only described in the Discussion section. The definition of RMSE was also not explained anywhere.

We moved this section from the Discussion to the Introduction, renamed it as “Model accuracy measure for avoiding superfitting: RMSE,” added description of the definition of RMSE, and adjusted wordings of the subsequent part of the Introduction to incorporate the difference between R2 and RMSE.

We hope this makes the manuscript convey the logical flow more effectively.

【Response to the comment from Reviewer2】

1. Comments to the Author

1. It is not true that the Whole-liver mass calculation problem is better done using simple metrics versus Deep-learning tools. These artificial intelligence tools have shown greater superiority in the field of prediction than statistical tools, although they require a greater inclusion of cases for training and subsequent validation. Although some of these classifiers are black-box models (for example, artificial neural networks) in which the predictor variables are not known, there are other classifiers such as Random Forest, which allow the variables and their relative risk to be known, especially in databases. small data sets, like the one used in this article.

(Our response)

We appreciate this thoughtful comment, which points to interesting recent advancement in Machine Learning (ML) research.

The primary author of this manuscript has first-hand experience in using ML models in clinical context, as listed below in case:

1. Inohara T, Ichihara N, et al. The effect of body weight in infants undergoing ventricular septal defect closure: A report from the Nationwide Japanese Congenital Surgical Database. J Thoracic Cardiovasc Surg. 2019;157(3):1132-41.e7.

2. Nishioka N, Ichihara N, et al. Body mass index as a tool for optimizing surgical care in coronary artery bypass grafting through understanding risks of specific complications. J Thoracic Cardiovasc Surg. 2019.

3. Matsuoka T, Ichihara N, et al. Antithrombotic drugs have a minimal effect on intraoperative blood loss during emergency surgery for generalized peritonitis: a nationwide retrospective cohort study in Japan. World J Emerg Surg. 2021;16(1):27. 

4. Ikawa F, Ichihara N, et al. Visualisation of the non-linear correlation between age and poor outcome in patients with aneurysmal subarachnoid haemorrhage. J Neurol Neurosurg Psychiatry. 2021;92:1173-80.

We agree that some ML models (we assume the “Deep-learning tools” here can be interpreted as ML models), e.g., Random Forest and XGBoost, may allow more accurate estimation in this context, if sufficient volume of observations are available. We also agree that their “black-box” nature can be practically overcome with interpretability tools, e.g., variable importance measures, SHAP, partial dependence plot, and LIME.

As described in the first paragraph of the Introduction, copied below, small volume of available data precluded use of ML models in this study. Also, we assumed simple linear formulas have their own use case different from ML models in this context.

Importance of simplicity in developing formulas for manual use

Despite recent advances in machine learning, there are still some clinical areas in which complex algorithms fail to provide clinically meaningful gain of accuracy compared to relatively simple formulas because of the limited availability of data and high variability of the subject, leaving traditional simple estimation/prediction formulas yet to be replaced. When developing a formula in such areas, in addition to predictive accuracy, simplicity and ease of use are major concerns because they are often used manually, and the complexity of the model might limit its use or lead to errors. 

Thus, we focused on applying the methodological features established in modern statistics/ML, other than ML models themselves, e.g., cross validation, to this area. We hope this is agreeable to the reviewer.

2. The sample size necessary for reliability is not specified. JLTS transplants since 2012 are analyzed to define the analysis cohort.

(Our response)

We appreciate this comment on the nature of the dataset we used for this study. Admittedly, the volume of data used for this study was determined by availability, not requirement of the modeling approach and precision. While we did not explicitly describe it because the same is the case with most observational studies based on registry data, we agree that preparing the “necessary” volume of data would allow a greater flexibility in selecting modeling approaches, e.g., application of ML models, with higher precision.

3. The results are only applicable to the database where the formula was created, since they do not include differential factors such as race, which can influence liver size.

(Our response)

We appreciate the reviewer’s comment. Please see our response to Reviewer #1’s comment #2.

---

## [Decision Letter · Decision Letter 1]

4 Jan 2023

Achieving clinically optimal balance between accuracy and simplicity of a formula for manual use: development of a simple formula for estimating liver graft weight with donor anthropometrics

PONE-D-21-33293R1

Dear Dr. Gotoh,

We’re pleased to inform you that your manuscript has been judged scientifically suitable for publication and will be formally accepted for publication once it meets all outstanding technical requirements.

Kind regards,

Sathishkumar V E

Academic Editor

PLOS ONE

Additional Editor Comments (optional):

Reviewers' comments:

Reviewer's Responses to Questions

**Comments to the Author**

1. If the authors have adequately addressed your comments raised in a previous round of review and you feel that this manuscript is now acceptable for publication, you may indicate that here to bypass the “Comments to the Author” section, enter your conflict of interest statement in the “Confidential to Editor” section, and submit your "Accept" recommendation.

Reviewer #1: All comments have been addressed

Reviewer #2: All comments have been addressed

2. Is the manuscript technically sound, and do the data support the conclusions?

Reviewer #1: Yes

Reviewer #2: Yes

3. Has the statistical analysis been performed appropriately and rigorously? 

Reviewer #1: Yes

Reviewer #2: Yes

4. Have the authors made all data underlying the findings in their manuscript fully available?

Reviewer #1: Yes

Reviewer #2: Yes

5. Is the manuscript presented in an intelligible fashion and written in standard English?

Reviewer #1: Yes

Reviewer #2: Yes

6. Review Comments to the Author

Reviewer #1: Well written and the strategy to achieve the balance between accuracy and simplicity is innovative, well explained and well executed

Reviewer #2: The work submitted by Ichihara et al is a manuscript that has resolved all the comments made by this reviewer. The limitations pointed out by the authors in their answers are convincing and typical of an analysis of these characteristics.

7. PLOS authors have the option to publish the peer review history of their article (what does this mean?). If published, this will include your full peer review and any attached files.

Reviewer #1: No

Reviewer #2: **Yes: **Javier Briceño

<quillbot-extension-portal></quillbot-extension-portal>

---

## [Editor Report · Acceptance letter]

11 Jan 2023

PONE-D-21-33293R1 

Achieving clinically optimal balance between accuracy and simplicity of a formula for manual use: development of a simple formula for estimating liver graft weight with donor anthropometrics 

Dear Dr. Gotoh:

I'm pleased to inform you that your manuscript has been deemed suitable for publication in PLOS ONE. Congratulations! Your manuscript is now with our production department. 

Kind regards, 

on behalf of

Dr. Sathishkumar V E 

Academic Editor

PLOS ONE